Effectiveness of physical activity on immunity markers and quality of life in cancer patient: a systematic review

AL-Mhanna Sameer Badri 1
Wan Ghazali Wan Syaheedah syaheeda@usm.my 1
Mohamed Mahaneem 1
Rabaan Ali A. 2
Santali Eman Y. 3
H. Alestad Jeehan 4
Santali Enas Y. 5
Arshad Sohaib 6
Ahmed Naveed 7
Afolabi Hafeez Abiola 8
1 Department of Physiology, School of Medical Sciences, Universiti Sains Malaysia , Kubang Kerian , Kelantan , Malaysia
2 Molecular Diagnostic Laboratory, Johns Hopkins Aramco Healthcare , Dhahran , Saudi Arabia
3 Department of Pharmaceutical Chemistry, College of Pharmacy, Taif University , Taif , Saudi Arabia
4 Immunology and Infectious Microbiology , Glasgow , United Kingdom
5 Pediatric Oncology Department, Maternity and Children Hospital , Makkah , Saudi Arabia
6 Periodontics Unit, School of Dental Sciences, Universiti Sains Malaysia , Kubang Kerian , Malaysia
7 Department of Medical Microbiology and Parasitology School of Medical Science, Universiti Sains Malaysia , Kubang Kerian , Malaysia
8 Department of General Surgery, School of Medical Sciences, Universiti Sains Malaysia , Kubang Kerian , Malaysia
Cè Emiliano
Electronic publication date: 2022 Aug 2
Publication date: 2022
Volume: 10
Electronic Location ID: e13664
Received 2021 Dec 28; Accepted 2022 Jun 10
Copyright: ©2022 AL-Mhanna et al.
Copyright year: 2022
Copyright holder: AL-Mhanna et al.
License: This is an open access article distributed under the terms of the Creative Commons Attribution License, which permits unrestricted use, distribution, reproduction and adaptation in any medium and for any purpose provided that it is properly attributed. For attribution, the original author(s), title, publication source (PeerJ) and either DOI or URL of the article must be cited.
License URL: https://creativecommons.org/licenses/by/4.0/

Keywords: Rehabilitation, Physiotherapy, Patient care, Palliative care, Interprofessional

Funding: Taif University Research TURSP-2020/330 This work was supported by Taif University Research Supporting project number (TURSP-2020/330). The funders had no role in study design, data collection and analysis, decision to publish, or preparation of the manuscript.

==============================
Background

Cancer is a huge group of diseases that can affect various body parts of humans but also has a psychological, societal, and economic impact. Physical activity can improve the quality of life (QOL) and immunity, while moderate intensity exercise can reduce the probability of this lethal disease. The current study aimed to determine the effect of physical activity on immune markers and QOL in cancer patients as well as to evaluate cancer-related fatigue (CRF) and its association with physical activity.

Methodology

Before starting the study, the study protocol was registered in PROSPERO (registration number CRD42021273292). An electronic literature search was performed by combining MeSH terminology and keywords used with the Boolean operators “OR” and “AND” to find relevant published studies on PubMed, Scopus, Cochrane, and ScienceDirect databases. The Joanna Briggs Institute (JBI) critical evaluation checklist was used to assess the quality of selected studies, while the GRADE approach was used to see the quality of evidence.

Results

A total of 13,931 studies were retrieved after the search on databases. After the scrutiny of studies by reading the title of articles and the inclusion/exclusion criteria, a total of 54 studies were selected for further screening by reading the full texts. In the final, a total of nine studies were selected for the current systematic review and proceeded for data extraction. The patients who were doing different exercises showed improvements in immunity, QOL, and reduction in CRF. A significant reduction in tumour necrosis factor-α (TNF-α), C reactive protein (CRP), interleukin-8 (IL-8), IL-6, and an increase in natural killer (NK) cells levels was also observed.

Conclusions

The exercise program is safe and beneficial to improve the quality of life and immunity markers before, during, and after cancer treatment. Physical exercise may also help patients to overcome the adverse effects of the treatment and to reduce the chance of developing new tumours in the future.

Introduction

Cancer is a vast group of diseases that can affect any part of the human body (Perez-Tejada et al., 2021). Other terms that can describe cancer are neoplasms and malignant tumours. A specific hallmark of cancer is the rapid development of anomalous cells that expand beyond their normal bounds, permitting them to infiltrate other body regions and expand to other tissues and organs; this is known as metastasis (Lanser et al., 2020). Cancer is a significant cause of death worldwide in 2020; and the most common cancers were lung, colon and rectum, liver, stomach, and breast cancers  (Ferlay et al., 2020; Hasham, Ahmed & Zeshan, 2020).

Physical activities can be explained as skeletal, muscular ambulation, or work that leads to energy consumption (Condello et al., 2016). Occupational, sports, conditioning, domestic and other types of physical activity can be seen throughout daily life. At the same time, exercise is a type of physical activity that is scheduled, structured, and done with the intention of improving or sustaining physical fitness as a final or intermediate aim. Physical exercise has been shown to enhance patients’ physical capacity and minimize illness and treatment-related adverse effects such as lower body strength, fatigue, QOL, and functional performance in patients with advanced-stage cancer (Albrecht & Taylor, 2012). Moreover, regular exercise helps to slow the progression of illness and decrease inflammation in people with chronic inflammatory disorders, and improve immunity markers (Shephard, 2017). Exercise regimens for breast cancer patients have been shown to impact the survival rates positively (Higgins et al., 2014). Increased physical exercise has been suggested a potential to significantly lower prostate cancer-specific and total death rates (Shephard, 2017). In addition to increased survival, many studies have shown an association between physical activity and improvement in QOL after a cancer diagnosis, including physical, functional, psychological, and emotional well-being  (Courneya & Friedenreich, 1999; Pinto, Dunsiger & Waldemore, 2013; Voskuil et al., 2010).

The World Health Organization states that exercise improves QOL, and moderate intensity exercise decreases the risk of cardiovascular disease, diabetes, and cancer. Exercise may positively impact cancer outcomes by lowering obesity and adipokine levels, lowering insulin, glucose, and sex hormone levels, increasing intestinal motility, lowering inflammation, and stimulating the immune system (Gleeson et al., 2011; McTiernan, 2008; Prieto-Hontoria et al., 2011). Cancer incidence and immune system dysfunction can also be as a result of age-related changes that can be mitigated to some extent by exercise  (Jones et al., 2009). The immune reaction can be reduced by partial inactivation of antigen-specific B and T cells, dendritic cells, neutrophils, macrophages, and NK cells (Senchina & Kohut, 2007; Bukhari et al., 2021). Therefore, pro- and anti-inflammatory cytokines are engaged in immune system regulation and play a critical role in cancer inflammatory processes (Loprinzi & Cardinal, 2013; Al-Hatamleh et al., 2022). Nevertheless, the molecular mechanisms that explain exercise multi-level effects in cancer patients are mainly unknown. Moreover, in rodents and cell culture research, many putative underlying mechanisms have been explored. However, human evidence is insufficient, and the clinically relevant cut-off points of the exercise-induced immuno-inflammatory response still require more investigation in cancer patients (Pedersen et al., 2016; Hojman et al., 2018). In addition, although it has been established that physical activity has anti-inflammatory benefits in healthy individuals, its anti-inflammatory effects in cancer patients suffering from CRF have not yet been identified (Hojan et al., 2016). Currently, it is unclear what kind of exercise is required to provide the best outcomes related to QOL and immune function (Brown et al., 2012b; Kruijsen-Jaarsma et al., 2013). The study aimed to determine the effect of physical activity on immune markers and QOL in cancer patients, and to evaluate CRF in cancer patients and its association with physical activity.

Materials and Methods

Protocols and registration

The study protocol was registered in PROSPERO with the registration number CRD42021273292. Before proceeding further with the systematic review and its registration in PROSPERO, the formal search using individual keywords against eligibility criteria was conducted in order to make sure the availability of enough studies to be included in the current review.

Research question

Studies about the effectiveness of physical activity on immunity and QOL amongst cancer patients were selected based on the “PICOS” (PRISMA-P 2016) technique:

“PICOS”

P (population) = Cancer patients

I (Intervention) = Exercise

C (Comparison) = Control

O (Outcome) = Improvement of immunity, QOL, and CRF

S (Study design) = Clinical studies

Search strategies

A few of the databases and search engines were used in this review as shown in Table S1.

Sources of data search

In order to validate the search strategy, two of the authors (S.A and H.A) have performed a literature search on PubMed, Scopus, Cochrane, and ScienceDirect on 15-02-2022 by combining the MeSH terminologies and keywords with the help of Boolean operators “OR” and “AND” to find out the relevant literature and to apply appropriate filters. The keywords “Neoplasm”, “-Malignancy”, “Immunology”, “Natural killer cells”, “Physical Activity”, “Exercise training” and “Quality of life” To obtain the required literature, the logical operations “OR” and “AND” were combined with the appropriate filters and the logical operators “OR” and “AND”. Table S1 shows the complete search method as well as the number of articles that resulted from it.

Criteria for eligibility of studies to be included

After a literature search, all studies that investigated the effects of physical activity on cancer patients’ immune markers, QOL, and CRF on cancer patients published till February 15th, 2022 were retrieved. Two authors (S.B and S.A) used the PICOS technique to examine the complete texts of the remaining publications and determine inclusion and exclusion criteria. In case of any conflicts, the third author (N.A) was approached until a final decision was reached.

Inclusion criteria

1. Studies that have physical activity/exercise intervention on cancer patients.

2. Diagnosed cancer patients with no age limit.

3. Publication in English with full text available.

4. Original clinical studies.

Exclusion criteria

1. All review articles, case reports, commentary, letters, and short communication.

Criteria for selection of studies

The selection of studies was done by two of the authors (S.B and S.A) based on inclusion and exclusion criteria, linear evaluation of titles, abstracts, and complete texts. In case of any discrepancies, a discussion with a third author (N.A) was done before reaching the decision of either inclusion or exclusion of articles for the current systematic review.

Data extraction

Two authors (S.B. and S.A.) performed independent sample data extraction from eligible studies after reading the entire text. The initial data extraction includes the first author’s name, journal name, population, year of publication, type of cancer, gender, method (exercise name, duration, intensity, sets, reps, timing of the intervention, duration of the study immunity and QOL parameter) outcomes and the conclusion of studies.

Quality assessment

The Joanna Briggs Institute (JBI) critical assessment checklist was used to appraise the quality of the included studies (Joanna Briggs Institute (JBI), 2017). This checklist assessed nine items: (i) appropriate sampling frame, (ii) proper sampling technique, (iii) adequate sample size, (iv) study subject and setting description, (v) sufficient data analysis, (vi) use of valid methods for the identified conditions, (vii) valid measurement for all the participants, (viii) using appropriate statistical analysis, and (ix) adequate response rate. Answers as yes, no, unclear, or not applicable are assigned to each item. The ‘yes’ response received a 1 score, whereas the ‘no’ and ‘unclear’ responses received 0 ratings. Finally, the average score for each item was computed. The quality of studies with scores below and above the mean was then classified as good and poor quality, respectively. The study was included or excluded based on the methodological quality assessment (Table S2).

Quality of evidence

The Cochrane Collaboration’s Grades of Recommendation, Assessment, Development, and Evaluation (GRADE) approach was used to assess the quality of evidence in included studies. The GRADE system provides four levels of quality, with randomized trial evidence being the highest. Depending on the existence of four elements, it might be degraded to moderate, low, or even extremely poor-quality evidence: (i) constraints in study design and implementation; (ii) indirectness of evidence; (iii) unexplained heterogeneity or inconsistency of results; and (iv) imprecision of outcomes. GRADEpro was used to represent the quality of evidence for each specific outcome (GRADEpro, 2015) (Table S3).

Results

Study selection results

A total of 13,931 studies were retrieved from PubMed, Scopus, Cochrane, and ScienceDirect using all MeSH keywords. After identifying duplicate articles (384), a total of 13,547 studies have proceeded for further selection. After reading the title and abstract of the articles, a total of 12,440 articles were excluded according as per general inclusion and exclusion criteria. The remaining 1,107 articles were proceeded for further selection by reading the full texts, out of which 1,053 were excluded. Data extraction was done from the nine studies that purely met the whole eligibility criteria (Fig. 1).

Figure 1 PRISMA flowchart for search strategy.

Study features

Tables 1 and 2 summarises the main features of the studies that were included in this systematic review. The studies selected were carried out in different countries during various time durations. Each of the included studies was published in a good, reputed journal. The technical characteristics were population, type of cancer, gender, type of exercise, duration of exercise, intensity, sets, repeats, the timing of the intervention, duration of the study immunity, and QOL parameters. In summary, in the included studies a total of 475 subjects were clinically assessed. The duration of each study was approximately 12 weeks. The studies were conducted on the head and neck cancers, breast cancer, hematologic cancers, small cell lung carcinoma, colorectal carcinoma, prostate carcinoma, liver carcinoma, cancer related to the bile duct, oesophageal cancer, and central nervous system, skin, urogenital, Hodgkin lymphoma and non-metastatic cancers. Cancer patients were introduced to the combination of different exercises like progressive AE and RT, RT alone, flexibility exercise, and rehabilitation program at home or hospital setup (Tables 1 and 2).

Table 1 General characteristics of included studies in the current systematic review.

Reference	Study design	Population	Name of journal	Year of publication	
Galvao et al. (2010)	RCT	Australian	Journal of Clinical Oncology	2010	
Hojan et al. (2016)	RCT	Poland	European Journal of Physical and Rehabilitation Medicine	2016	
Pierce et al. (2009)	RCT	Poland	Polish Archives of Internal Medicine	2017	
Chasen et al. (2013)	******	Canada	Current Oncology	2013	
Sprod et al. (2010)	Pilot study	America	Community Oncology	2010	
Kim et al. (2018)	Experimental study	Korea	Thyroid	2018	
Ergun et al. (2013)	RCT	Turkey	European Journal of Cancer Care	2013	
Baumann et al. (2010)	RCT	Germany	Bone Marrow Transplantation	2010	
Parent-Roberge et al. (2020)	Pilot study, A single-blind, two arms RCT	Canada	Brain, Behaviour, and Immunity-Health	2020	

Table 2 The technical characteristics of included studies in the current systematic review.

Reference	Sample size	Cancer type	Gender	Method (exercise name, duration, intensity, sets, reps).	Timing of intervention	Duration of the study	Immunity and quality of life parameter or immune component examined	Results	Conclusion	
 Perez-Tejada et al. (2021)	57
EX = 29
Co = 28	Prostate	Male	Patients undergo combined progressive AE & RT.	More than 2 MO after starting with androgen deprivation therapy	12 W	General health; QLQ-C30 and C-RP	Significant improvement in QOL: pre-EX: 66.0 ± 23.1, post-EX: 71.4 ± 17.5, P-value = 0.02.	Improve well-being and overall QOL of patients. The CRP significantly decreased following EX.	
				RT included (upper- and lower-body 8 EX, 2-4 sets, 12-6RM) & AE (Progressive cycling and walking 2/w) 15–20 M, 65–80%				CRP: pre-EX: 2.7 ± 3.2, post-EX: 1.8 ± 1.1, P-value = 0.008		
				HR peak and 11-13 RPE						
 Lanser et al. (2020)	54
EX = 27
CO = 27
	Prostate	Male	The intervention group undergoes AE for 30 M and 15 M of RT (sets of 8 reps at 70% to 75% of their estimated 1RM). Both EX was about 50 to 55 M at 65–70% VO2M for 5/D/W. The CO group revised only usual care.	EX started one W before radiotherapy	8 W	TNF- α, pro-inflammatory cytokines (interleukin IL-1 β, IL-6) F, and QOL	Significantly decreased pro-inflammatory cytokine levels: IL-6: pre-EX: 43461.35 ± 1824.8, post-EX: 907.5 + 2460.7, TNF- α: pre-EX: 50.01 ± 20.63, post-EX: 61.74 + 39.12, and IL-1 β: pre-EX: 98.57 ± 180.12, post-EX: 96.72 ± 134.03, P-value = < 0.05.	Improves inflammatory markers (by decreasing their levels), decreases F, and improves QOL in high-risk PC patients after radiation therapy.	
								Decreased F: pre-EX: 42.7 ± 2.1, post-EX: 43.9 ± 5.0, P-value = < 0.05		
								Improved QOL: pre-EX: 70.7 ± 2.1, post-EX: 72.3 ± 6.3, P-value = < 0.05		
 Ferlay et al. (2020)	66
EX = 35
CO = 31	Prostate	Male	The intervention groups undergo AE; 5/S/W for 8W and 3/D/W for the next 10 MO for 30 M.	During and after radiotherapy and androgen deprivation therapy	12-M0	Inflammatory factors, FACT-G score, and QLQ-C30	Improve all the aspects of QOL: pre-EX: 70.7 ± 2.1, post-EX: 65.91 ± 4.8, P-value = 0.001.	EX improved QOL and reduce pro-inflammatory cytokine levels in patients with PC undergoing radiotherapy and androgen deprivation therapy.	
				& RT; (2 sets of 8 reps at 70% to 75% for 25 M of their estimated 1RM). Both EX was about 65 to 70 M. with 65% to 70% VO2M. After radiation therapy, the intervention group did the same program, 3/W, but 1.5 H/D for 40 M of AE & 35M of RT. with 70% to 80% of heart rate reserve. The CO received only usual care				F: Pre: 42.7 ± 2.1, Post: 39.8 ± 3.7, p-value = 0.001.		
								No significant difference in IL- 1: pre-EX:106.6 ± 226.6, post-EX: 150.6 ± 1933.8, P-value = 0.18, and TNF- α levels: Pre-EX: 32.8 + 161.1, Post-EX: 84.6 ± 262.7, P-value = 0.71		
								Significant difference in IL ¬ 6: Pre-EX: 3158.1 ± 1675.2 and Post-EX: 150.6 ± 1933.8, P-value = < 0.001.		
 Hasham, Ahmed & Zeshan (2020)	67	Head and neck, Breast, Hematologic =, Non-small cell lung Small cell lung, Colorectal, Prostate, Liverbile duct , Esophageal Central nervous system, Skin, Unknown primary, Urogenital, Hodgkin lymphoma	Male and female	67 patients engaged in Rehab program for 8 W at the hospital gym 2D/W under the direction of an Occupational Therapist	After anticancer therapy	8w	Esas, CRP, F and QOL	A significant relationship between normal serum CRP and program completion, pre-EX: 3.15 ± 2.97, post-EX: 3.68 ± 3.23, P-value = 0.02.	Patients with different advanced types of cancers showed considerable improvements in functioning and QOL across several categories. The study concluded that an average amount of C-RP could indicate program completion.	
								Significant improvement in QOL: pre-EX: 4.85 ± 2.62, post-EX: 3.89 ± 2.41, P-value = 0.01.		
								F: pre-EX: 4.89 ± 2.6, post-EX: 3.81 ± 2.26, P-value = 0.001.		
 Condello et al. (2016)	38
EX = 19
CO = 19	Prostate and breast	Male and female	AE: (walking) increasing with 5–20% of steps at 3–5 RPE & RT: 11 EX, increasing towards 4 sets 15 reps: Low to moderate intensity. Patients were engaged in daily EX.	Following primary diagnosis, starting radiotherapy of at least 6 W	4W	PSQI scores, IL-6, and sTNF-R	Improvement in sleep quality: pre-EX: 7.06 ± 4.26, post-EX: 6.00 ± 3.87, P-value = 0.37.	Changes in sleep measurements and inflammatory markers were not associated with the intervention group. The relationship was found between TNF- α and subjective sleep quality as well as TNF- α sleep delay, patients had a better subjective sleep quality, and sleep efficiency was linked with higher levels of sTNF-R and IL-6p, respectively.	
								IL-6: pre-EX: 1.08 (range 0.06–2.98), post-EX: 1.38 (range 0.29–6.41) were increased, P-value = 0.31. While sTNF-R was decreased: pre-EX: 760.62 (range 448.64–1,476.21), post-EX: 680.52 (range 361.68–1,319.53). P-value = 0.59.		
 Albrecht & Taylor (2012)	43, EX= 22
CO=21	Thyroid cancer	Male and female	Patents were asked to perform AE (walking, 3–5 D/W for at least 150 M/W), RT (upper body EX for 2/W for more than 2 sets/ times) & flexibility EX (5 M before and after AE and RT).	During taking thyroid hormone medicine	12 W	HADS-A, EORTC QLQ-C30, NKCA and NKCC	Anxiety: pre-EX: 13.86 ± 3.31, post-EX: 11.32 ± 2.59, and F: pre-EX: 4.48 ± 1.46, post-EX: 3.52 ± 1.74 were significantly decrease, P-value = 0.001.	In thyroid cancer patients, a home-based EX program is beneficial in lowering F and anxiety, enhancing QOL, and boosting immunological function. A home-based EX program can be implemented for cancer patients.	
				30–40 M of EX were considered appropriate.				Improvement in QOL: pre-EX: 70.51 ± 12.33, post-EX: 82.73 ± 10.49, P-value = 0.001.		
								NKCA cytotoxicity: pre-EX:11.09 ± 7.71, post-EX: 14.46 ± 8.28, and NKCC cell: pre-EX: 10.93 ± 5.05, post-EX: 12.65 ± 5.86, in the intervention group were significantly increased, P-value = 0.004.		
 Shephard (2017)	60	Breast	Female	Patients were divided into three groups: 1—supervised EX group (AE + RT + education programme, n = 20); performed 45 M/D for 3 D/W under the direction of a professional doctor. 2—Home EX group (AE + education programme, n = 20); performed brisk walking for 30 M/D for 3 A/W. Groups 1 and 2 undergo 12 W of EX. 3—education programme (n = 20) followed up for 12 W.	Following breast cancer treatments.	12 W	IL-8, TNF- α, ENA-78, EORTC QLQ-C30, BFI and BDI.	Post-treatment IL-8: pre-EX: 10.37 ± 3.60, post-EX: 7.76 ± 3.10, and ENA-78: pre-EX: l9.64 ± 3.81, post-EX: 7.34 ± 5.29, and TNF- α: pre-EX: 13.01 ± 6.72, post-EX: 11.60 ± 4.71 levels were significantly decreased in the home EX group P-value = 0.03. BDI: pre-EX: 7.75 ± 6.69, post-EX: 4.70 ± 4.10, and BFI: pre-EX: 3.44 ± 2.23, post-EX: 2.86 ± 2.02, and QOL: pre-EX: 80.35 ± 11.22, post-EX: 85.67 ± 8.07, were significantly decreased following the EX, P-value = 0.004.	EX-induced changes in angiogenesis and apoptosis-related molecules imply that these parameters may be affected by EX. Patients with breast cancer who have completed their treatments have improved their QOL and reduced their depression.	
 Higgins et al. (2014)	64
EX = 32
CO = 32	Malignant with the indication haematopoietic	Male and female	EX group undergo aerobic endurance training 2/D on a bicycle ergometer 10–20 M during hospitalization & activities of daily living include walking on the hospital’s corridor every D (except weekend). CO group undergoes 20 M/D under the professional therapist at all times.	Undergoing haematopoietic stem cell transplantation	Approximately 6 W	EORTC QLQ-C30, haematological parameters (leucocytes, plts, Hb)	All haematological measures show significant declines in the intervention group: leucocytes: pre-EX: 6.7 ± 6.1, post-EX: 4.0 ± 1.6, Plts: pre-EX: 157.8 ± 98.1, post-EX: 83.6 ± 69.5, and Hb: pre-EX: 10.8 ± 2.2, post-EX: 9.4 ± 1.4, P-value = 0.001. Significant differences in favour of the EX-group concerning the QOL: pre-EX: 75.8 ± 21.8, post-EX: 61.6 ± 22.7, P-value = 0.006.	EX might have some favourable impacts on the patient’s physiological, psychological, and psychosocial levels during the full duration of the Hematopoietic stem cell transplant. During the EX-period, the majority of the patients were found to have steady neutrophil engraftment. Patients who underwent hematopoietic stem cell transplantation were not exposed to any additional risks, and on the contrary, the training program appeared to have aided in the patient’s recovery process.	
 Courneya & Friedenreich (1999)	20
EX = 10
CO = 10	Non-metastatic cancer		EX group undergo combined AE and RT. AE was 20 to 40 M (+5 M of warm-up and 5 M cool-down with 50% to 75%) intensity of estimated heart rate reserve (VO2M) & RT intensity was from 1 set of 12–15 reps to 2–3 sets of 10–15 RM. Patients participated 3/S/W, with 2/S under the supervision of the EX-physiologists. While the CO performed 2/S/W of supervised static stretching 30–45/M/S	During chemotherapy	12W	Inflammatory profile pro-inflammatory cytokines (IL-6 and IL-1 β), anti-inflammatory cytokines (IL-10, IL-1ra, and IL-15), and CRP. FACIT-F and PASE questionnaire	The intervention group showed no deterioration of the inflammatory profile and cancer-related F following the intervention, IL-1 β: pre-EX: 0.9 (0.4–1.1), post-EX: 1 (0.4–1.6), IL-10: pre-EX: 1.7 (0.5–9.2), post-EX: 8.2 (4.5–16.4), IL-15: pre-EX: 2.0 (1.8–4.6), post-EX: 2.5 (1.8–6.6), and IL-6: pre-EX: 0.7 (0.2–3.7), post-EX: 0.5 (0.1–3.3). P-value = 0.40. A decline in the CRP: pre-EX: 4.7 (3.8–5.8), post-EX: 3.1 (2.0–8.1) was observed in the intervention group. associations reaching significance were observed for the delay in D post-intervention in IL-1 β, IL-15, and leptin. A lower IL-6/IL-10 ratio: pre-EX: 0.47 (0.3–0.67), post-EX: 0.73 (0.18–3.2) in the intervention group was also found, P-value = 0.03. There was a trend for a significant increase of 4 points on the FACIT-F scale in the intervention group scores following the EX, P-value = 0.10.	The study concluded that, in breast cancer patients undergoing chemotherapy, no rise in pro-inflammatory markers was shown, EX had a favourable effect on cancer-related F and pain. EX might be a positive factor to improve QOL and decreased F. For cancer patients in the early stages of therapy, combined EX training appears to have a beneficial effect on cancer-related F without altering the fasting systemic pro-inflammatory profile.	
Notes.

EX exercise

CO control

MO Month

RT Resistance training

AE Aerobic exercise

W Weeks

M Minute

VO2M Maximal heart rate

S Session

H Hour

D Day

The Edmonton Symptom assessment system = RM Repetition maximum

RPE Rating of perceived

FACT-F Functional Assessment of Cancer Therapy–Fatigue

FACT-G Functional Assessment of Cancer Therapy-General

EORTC QLQ-C30 European Organization for the Research and Treatment of Cancer Quality of Life Questionnaire

HADS-A Hospital Anxiety-Depression Scale-Anxiety

NKCA natural killer cell activity

NKC natural killer cell

ENA-78 epithelial neutrophil activating protein 78

VEGF vascular endothelial growth factor

GRO growth related oncogene alpha

RANTES regulated upon activation, normal T-cell expressed, and secreted

ANG angiogenin

PDGF platelet derived growth factor

TRO thrombopoetin

MCP 1 monocyte chemotactic protein-1

MCP2 monocyte chemotactic protein-2

MCP3 monocyte chemotactic protein-3

BFI the brief fatigue inventory

BDI Beck Depression Inventory

CRP C-reactive protein

TNF- α tumour necrosis factor alpha

F fatigue

Esas the Edmonton Symptom Assessment System

The result showed that combined progressive aerobic exercise (AE) and resistance training (RT) in prostate cancer patients undergoing androgen suppression therapy showed significant improvement in QOL, decreased tiredness, and lower C-reactive protein (CRP) levels (Galvao et al., 2010). Combined progressive AE and RT in prostate cancer patients one week before radiotherapy, revealed a significant decrease in pro-inflammatory cytokine levels, fatigue, and improved QOL (Hojan et al., 2016). Combined progressive AE and RT in prostate cancer patients during and after radiotherapy and androgen deprivation therapy reduced pro-inflammatory cytokine levels and improved QOL, fatigue, and TNF-α levels. There was no significant difference in TNF-α, IL-1, and IL-6 levels (Hojan et al., 2017). Rehabilitation program for deafferents types of cancer after anticancer therapy revealed significant improvements in CRP level, QOL, and fatigue (Table 2)  (Chasen et al., 2013). Combined AE and RT improved QOL in patients with prostate and breast cancer who were undergoing radiotherapy (Sprod et al., 2010). Another study reported that, AE, RT, and flexibility exercise in thyroid cancer patients who are taking thyroid hormone medicine significantly decrease fatigue and anxiety, improve QOL, and significantly increase NK cells  (Kim et al., 2018). After breast cancer treatments, AE and RT resulted in significant reductions in IL-8 and an increase in Monocyte chemotactic protein-1 levels (Ergun et al., 2013). Aerobic endurance training in cancer patients undergoing haematopoietic stem cell transplantation showed significant improvement in QOL (Baumann et al., 2010). Combined AE and RT in non-metastatic cancer patients during chemotherapy, a decline in the CRP was observed in the intervention group. Associations reaching significance were observed in IL-1, IL-15, and leptin. A lower IL-6/IL-10 ratio and an improvement in QOL (Parent-Roberge et al., 2020).

The duration of exercise varies in different studies ranging from 15 to 70 minutes/ session. The improvement in the immunity observed in blood samples of the patients doing exercise at home revealed a significant reduction in TNF-α, CRP, IL-8, and an increase in NK cell levels which ultimately give advantages to the cancer patients. Furthermore, questionnaires on QOL reflects a reduction in CRF and improvement in the QOL among patients who were doing regular exercise. The technical characteristics of the studies have been shown in Table 2.

Quality of the evidence

The quality of evidence of the included studies was very low certainty. For most studies in most domains, there was a low risk of bias. There was no evidence of selective reporting bias and the performance bias was judged to have a low risk of bias. There was considerable heterogeneity, no change in the impact estimate was seen, and the result remained significant even though the 95% CI was broader in all cases. The overall level of evidence contributing to this review as assessed using the GRADE approach was a considerable quality.

Discussion

The study aimed to determine the effect of physical activity on immune markers and QOL in cancer patients. Also, to evaluate CRF and its association with physical activity. This review observed significant improvements in QOL, fatigue, and immunity markers such as NK cells and other cytokine profiles following exercise. The presence of NK cells in the body confers immunity that helps to fend or kill cancer cells by secreting perforins and granzymes (Zuo & Zhao, 2021). Upon contact with cancerous cells, NK cells form immune synapses to deliver the lethal hit. These findings are consistent with those of other studies involving a well-trained study population (Kruijsen-Jaarsma et al., 2013; Kawada et al., 2010) and asthma patients (McFarlin, Hutchison & Kueht, 2008). These significant findings could be related to the different types of exercise performed by different cancer patients, indicating that immunity of the cancer patients can be increased with many types of exercise.

TNF-α levels are linked to a wide range of diseases and chronic inflammatory processes and have been associated with delayed wound healing (Ashcroft et al., 2012). The outcome of the present study showed that TNF-α, as well as other inflammatory markers such as IL-6 and CRP, were found to be significantly reduced following exercise intervention. This was likely due to the anti-inflammatory effects of the physical activity performed by the exercise group (Petersen & Pedersen, 2005). IL-6 has been associated with fatigue in breast cancer survivors (Schubert et al., 2007; Saligan & Kim, 2012), the most prevalent and debilitating symptom among cancer survivors, and some studies showed that physical activity reduces fatigue in people with breast cancer specifically (Battaglini et al., 2014; Zou et al., 2014), and people with cancer in general (Meneses-Echávez, González-Jiménez & Ramírez-Vélez, 2015; Meneses-Echavez, Gonzalez-Jimenez & Ramirez-Velez, 2015). In addition to being linked to fatigue, IL-6 has been shown to predict survival in patients with breast cancer (Salgado et al., 2003). Therefore, this result can help in understanding the positive trend in survival due to physical activity in various cancer populations. Indeed, since chronic inflammation is well known to play a critical role in cancer progression, development, and survival, the finding of decreases in a variety of cytokines (particularly IL-6, and TNF-α) may have similar implications (Coussens & Werb, 2002; Pierce et al., 2009). Previous studies comparing long-term yoga practitioners to non-yoga practitioners found that long-term practitioners had significantly lower levels of TNF-α, CRP, and IL-6 than non-yoga practitioners (Kiecolt-Glaser, 2010; Vijayaraghava et al., 2015). Huffman et al. (2008) concluded that exercise training reduces the expression of some proinflammatory cytokines, including TNF-α and IL-6, by reducing adipose tissue. These findings suggest that consistent reductions in circulating pro-inflammatory markers can be achieved through regular exercise.

Early integration of palliative care in the course of disease for patients with incurable cancers has gradually drawn interest as a viable and effective strategy not only to enhance QOL and mood, but also to prolong survival (Shephard, 2017; Thornton, Andersen & Carson, 2008). According to Hojan et al. (2016), frequent, moderate-intensity exercise improves functional ability, reduces inflammatory markers and fatigue, and improves QOL in high-risk prostate cancer patients undergoing radiation. Similarly, Kim et al. (2018) found that a home-based exercise-program can help patients receiving thyroid hormone replacement after thyroid surgery reduce fatigue and anxiety, improve QOL, and boost immunological function. Similar previous studies on different cancer subjects also reported that the exercise-programs are safe and helpful in improving QOL and reducing depression in cancer patients who have completed their treatments (Spence, Heesch & Brown, 2010; Pacheco et al., 2019; Battaglini et al., 2009; Daly et al., 2020; Williams et al., 2021).

Ergun et al. (2013) conducted research looking at the effects of exercise on angiogenesis and apoptosis-related markers, QOL, fatigue, and depression in breast cancer survivors. In that study, 63 patients with breast carcinoma were divided into three groups: supervised exercise, home exercise, and education. It was found that IL-8 and neutrophil-activating protein-78 levels in the home and the supervised exercise group were substantially lower after the exercise-program. Monocyte chemoattractant protein-1 levels increased significantly in the education group. Depression and fatigue were significantly lower after the intervention in the supervised exercise group. Both home and supervised exercise groups showed improvement in QOL after the exercise-programs, while no differences were reported in the education group. This might be due to the use of various quality of life scales and the variances in the type, duration, and intensity of exercise  (Demark-Wahnefried et al., 2006; Cadmus et al., 2009).

Cancer rehabilitation is a procedure that helps people work within their physical, social, psychological, and vocational limitations  (Basen-Engquist et al., 2006; Cho, Yoo & Kim, 2006). Physical symptoms (fatigue and physical endurance), nutritional symptoms (poor appetite, unintentional weight loss, and nutritional deterioration), psychological symptoms (anxiety, depression, and nervousness), and overall QOL have been shown to improve with post-rehabilitation (Pedersen et al., 2016; Hojan et al., 2016; Guinan, Connolly & Hussey, 2013). In a previous study, patients who had undergone a cancer rehabilitation program qualitatively said that rehabilitation was a significant stepping stone in their recovery and that physical and psychological treatment was an important mix (Cho, Yoo & Kim, 2006). Apart from the advantages mentioned above, the physical activity results differ depending on the stage of the disease or the research patients with incurable advanced disease (palliative care patients) (Lanser et al., 2020; Mustian et al., 2009). In patients with prostate cancer undergoing radiation, long-term supervised exercise training is more beneficial than educational materials on physical activity to lower cardiovascular risk and improve functional status (Hojman et al., 2018; Hojan et al., 2017).

According to the systematic studies, exercise enhanced QOL in cancer patients with significant evidence, particularly in breast cancer  (Perez-Tejada et al., 2021; Boivin et al., 2020; Inglis et al., 2021). The QOL scores in exercising cancer patients were shown to be higher when compared to controls or baseline values (Husson et al., 2013; Sadeghi et al., 2020). Some studies demonstrate exercise does not affect QOL in breast cancer patients who have completed the treatment (Basen-Engquist et al., 2006; Benoy et al., 2002; Brown et al., 2012a; Burnham & Wilcox, 2002). This might be due to the use of various QOL scales/inventory in studies, the multifactorial nature of QOL, variances in the type, length, and intensity of the exercise, a scarcity of high-quality research, or discrepancies in study populations. After the exercise program, the functional and general health scores of the EORTC QOL-C30 in the supervised exercise group and the functional score of the EORTC QOL-C30 in the home-exercise group were significantly improved (Cramp & Byron-Daniel, 2012). In similar observational research, individuals with a lengthy period following diagnosis had better baseline function and QOL scores (Daly et al., 2020).

Fatigue is a common early symptom of cancer that can develop before therapy and worsen throughout treatment (Jones et al., 2009). Fatigue affects almost all patients, with a reported frequency of 99% (DeSantis et al., 2015). While some studies have shown that walking, AE, and resistive exercise-programs can reduce CRF (Williams et al., 2021; Inglis et al., 2021; Husson et al., 2013). Physical activity may help to lower CRF by activating neuromuscular function and causing hemodynamic changes, as well as by decreasing social isolation (Schubert et al., 2007; Bower et al., 2000). However, results from Ji et al., exhibited that there was no significant effect of the exercise programs on CRF (Lee et al., 2010). According to  Parent-Roberge et al. (2020), exercise may have had a beneficial effect on CRF that was not mediated by changes in the pro-inflammatory cytokine profile.

To examine the process of changes in QOL, fatigue, and depression in cancer patients in response to different exercise programs, requires a larger scale survey. The beneficial effect of physical activity on cancer therapy can improve systemic immunity and cancer prognosis through adequately structured physical activity; thus, this would be an excellent public-health intervention for reducing the impact of cancer at relatively low cost and risk. Psychosocial support can also be another relevant topic to be investigated in future studies. Finally, the longitudinal follow-up should be planned to determine whether better QOL would help to improve the severity of diseases in patients or not.

Study Limitations

Only those studies included in the present systematic review that published in the English language which may result in the leftover of some studies. Furthermore, the case reports, short communications, and conference papers were also excluded. Another limitation of the current study was the heterogeneity of included studies, because of which it is difficult to say anything about one specific type of cancer. Apart from this, there was no such limitation for the review process.

Conclusions

From the outcome of the present study, it was found that physical exercise can reduce cancer incidence by improving immunity markers such as TNF-α, IL-6, IL-10, and NK cells. However, only through an indebted understanding of the impact of physical activity on the growth and mechanisms of tumour cells, metabolic pathways, and systemic immune functions might help to fully understand the correlation between exercise, cancer prevention, and treatment. In the future, this may also enable adequate administration of personalized exercise treatment plans for patients with cancer. However, no single study can solely identify a particular form of exercise as the most prudent or futile approach to improving the QOL, fatigue, and immunity markers in cancer patients. Hence, investigations on the effects of various forms of exercise on QOL and immunity in cancer patients are required.

Supplemental Information

File S1 Formal search strategy and Quality assessment raw data

Click here for additional data file.

File S2 Study Justification and rationale of current systematic review

Click here for additional data file.

Supplemental Information 1 PRISMA checklist

Click here for additional data file.

Data S1 Raw Data

Click here for additional data file.

Additional Information and Declarations

Competing Interests

Author Contributions

Data Availability

The authors declare there are no competing interests.

Sameer Badri AL-Mhanna conceived and designed the experiments, prepared figures and/or tables, and approved the final draft.

Wan Syaheedah Wan Ghazali conceived and designed the experiments, prepared figures and/or tables, and approved the final draft.

Mahaneem Mohamed analyzed the data, authored or reviewed drafts of the article, and approved the final draft.

Ali A. Rabaan analyzed the data, authored or reviewed drafts of the article, and approved the final draft.

Eman Y. Santali analyzed the data, authored or reviewed drafts of the article, and approved the final draft.

Jeehan H. Alestad performed the experiments, authored or reviewed drafts of the article, and approved the final draft.

Enas Y. Santali performed the experiments, authored or reviewed drafts of the article, and approved the final draft.

Sohaib Arshad conceived and designed the experiments, prepared figures and/or tables, and approved the final draft.

Naveed Ahmed conceived and designed the experiments, prepared figures and/or tables, and approved the final draft.

Hafeez Abiola Afolabi performed the experiments, prepared figures and/or tables, and approved the final draft.

The following information was supplied regarding data availability:

The raw data is available in the Supplemental File.

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
