# Peer review of "Effectiveness of physical activity on immunity markers and quality of life in cancer patient: a systematic review"

_PeerJ, doi:10.7717/peerj.13664_

## Round 0.1 · original submission · Major Revisions

· Academic Editor

Major Revisions

Dear Authors three experts in the field revised your manuscript retrieving some major points you should address while reviewing your paper.

Reviewer 1 ·

Basic reporting

No comment

Experimental design

The authors investigated few databases. It is mandatory to investigate Web of Science and EMBASE.

It is also mandatory to investigate the gray literature.

The authors do not assess the certainty of evidence with GRADE. It is currently considered a mandatory and necessary step in systematic reviews with or without meta-analysis.

Validity of the findings

The few databases investigated leave doubts as to whether all the articles were retrieved in the search.

Without GRADE, doubts persist as to the certainty of evidence of the selected articles.

Additional comments

It is necessary to review the search strategy, investigate other databases and gray literature, and assess the certainty of evidence by GRADE.

Reviewer 2 ·

Basic reporting

- The spelling and language overall are not great. You put capital letters where it is not needed, you make grammatical errors, you use brackets but never close the bracket, …I suggest that you check your manuscript for these little errors and let someone who is proficient in English review the manuscript.


Abstract:
- Your abstract is very dense and detailed, the aim and the most important aspects of this review are not clear.

Introduction:
- There is no need for figure 1 or 2, they do not add anything to your manuscript.
- Too much information that is not relevant for the reader: the most prevalent cancers with their numbers don’t need to be described. You can speak on this as a more general description, but this is too detailed and not needed.
- Reference 9: you use this reference when you talk about patients in general, however, this reference is of a study on prostate cancer. This reference is not relevant. This is a general remark: use references that are relevant to your review fe. all cancer patients, as your review is also on general cancer patients.
- In your introduction, it should be clear what you mean when you talk about cancer patients. Is this anyone who has a diagnosis or is in active treatment, those who are cancer survivors?
- Line 90-92: So QoL is not determined by cardiovascular fitness or exercise capacity but only by sleep quality and feeling empowered?
- The evidence of exercise and physical activity on cancer-related symptoms is vast, I want to see better references to this. Even on recurrence rate, survival rate, … The importance of exercise is not really clear from your introduction
- Line 112-115: So you will only look at cancer patients suffering from cancer-related tiredness?
- The aim of your review remains very unclear after your introduction
- Your introduction right now is just a summation of facts, I miss a comprehensive introduction. Use the problem-gap-hook method.

Figures & Tables
- Figures 1 & 2 are not necessary
- Table 4: you report on different outcomes than the ones you are interested in: fatigue, anxiety, sleep, strength, …

Experimental design

What is your actual research question? You mention cancer patients suffering from cancer-related tiredness, but then you don't mention this anywhere else?

Methods:
- I (Intervention) = Exercise and QOL : how is QoL an intervention?
- Why do you include these specific journals, are they not available through pubmed, scopus search?
- Line 155-156: Why only between 2010-2020 ?
- Line 182-183: If two reviewers could agree? How did you agree? Did you not use the scoring results as a guide on whether to in or exclude the study based on methodological quality?
- In and exclusion criteria are very broad for example: What are cancer patients? is this anyone with a diagnosis, anyone with curative treatment, those with a terminal diagnosis, those who survived cancer and are post-treatment,… ?
If you include all these type of patients in your review, you have too much heterogeneity
- High-intensity exercise is an exclusion criteria, why?

Results:
- Line 187-190: You retrieved another 287 articles by hand searching? This is an indication that your search strategy was not good and that in fact you haven’t really conducted a proper systematic review
- Line 207: if you only looked at studies between 2010-2020 then it is obvious you will only find those articles.
- You describe in detail the journals from which you retrieved the articles (line 197-207) but you do not do these when it comes to the actual relevant information. I want to know more about the content and not which journal it was published in. What type of exercise has an influence on what type of immunity markers, what type of patients benefit of this exercise? Make a synthesis of your findings in the result section.

Validity of the findings

You results: you only describe these results in table 4 but do you have the data on this (p-values, ...)?

Discussion:
Your discussion has no common thread. You might check some of prisma statements but I miss an in-depth discussion. I do not find any discussion on your own findings, potential mechanisms why exercise could have these results, recommendations, critical remarks, …

- Line 253-260: It was found that IL-8 and neutrophil-activating protein-78 levels in the home-exercise group were substantially lower after the exercise-programme --> but not in the supervised exercise group: how do you explain this?
- line 292-294: I don’t understand the relevance of this sentence, what do you mean? Is this a recommendation for future research, what medication?

Additional comments

I have read the manuscript with great interest on this very relevant topic.
However, this review lacks quality. My most important remark is that you incorporate a lot of aspects but are never really to the point. Therefore the review is often incomprehensive. The description of the methods used is insufficient, the reporting of the results is unclear and the discussion section is not strong and does not add to the manuscript. This manuscript would benefit from an extensive revision before being considered further for publication.
I composed my more detailed remarks in the sections and gave some suggestions on how you can improve this manuscript.

·

Basic reporting

General comments
The study carries an appreciable novelty, considering the continuously increasing incidences of cancer on a global scale. The authors have ingeniously considered reviewing the effect of physical activity as a panacea to the cancer issue. However, there are several concerns with the manuscript, and a major revision is needful.

Technical arrangement and accuracy
The construction of several sections of the manuscript should be reordered for soundness and technical comprehension. For example, Lines 35-36 reads “ A formal search of the keywords was performed before proceeding with THE REGISTRATION AND FURTHER for the systematic review in order to make sure the availability of enough…” This statement, though somewhat valid, sounds quite twisted and might be difficult to comprehend. A revision might improve it.
Authors stated in lines 61-62 “Cervical cancer is one of the most prevalent cancers in females…” Then, in the latter parts, precisely lines 69-70, it was stated that “On the other hand, Breast carcinoma is REGARDED AS A THE MOST frequent medical illness that women face across the world”. The statement in lines 69-70 is both grammatically incorrect and inconsistent to the fact stated in lines 61-62. Furthermore, if other cancers peculiar to women would be mentioned in the introduction, it might be suggestable for them to follow one path. They shouldn’t be mentioned in a paragraph in lines 61-62 and repeated without any different point in lines 69. They should be ordered in one place in the introduction section.
References were made to GCO (the Global Cancer Observatory (fmr GLOBOCAN)) in lines 62-63, and the citation was not offered. Though the citation was apparently offered in the next sentence (line 65), but it would technically suffice if the source that mentioned GCO was mentioned in line 63
In line 73, it was mentioned that “New Zealand than in INTERMEDIATE ADVANCED NATIONS such as Europe (central)…” There is no technical classification of any set of nations as intermediate-advanced, if there is, the authors can provide a reference for this. The conventional classification, according to world bank, is based on Gross National Income per capita as low income countries (similar to least developed countries), lower-middle income countries, upper-middle income countries, high income countries (similar to developed countries) (https://blogs.worldbank.org/opendata/new-country-classifications-income-level-2019-2020). This should be revised.
In line 79-80, it was stated that “Physical activities can be explained as skeletal, muscular ambulation or work that leads in energy dissipation”. An explanation as this should have a recent and citable reference literature.
Authors declared in lines 103-104 that “Cancer incidence and immune system dysfunction ARE ALSO age-related changes that can be mitigated to some extent by exercise”. This claim might be contentious, as young people in some populations have been diagnosed with cancer and immune issues. “ARE ALSO” can be suggestively substituted with “can also be as a result of…”
In line 108, the word EXERCISE was capitalized. This should be reviewed if such punctuation has no significance to the context.
Study rationale
This section is seemingly wrongly positioned and of little technicality. It looks like a declaration of results “We found that…” It is, as other several parts of the manuscript, replete with the use of the pronoun WE which should not be used for technical writing.
Line 119 states “effects. we found that Natural Killer cytotoxic activity increased after…” This line has issues related to 1. wrong capitalization for the commencement of a fresh sentence 2. The use of the pronoun “we” 3. An undefined term “Natural Killer” introduced with questionable capitalization 4. “Natural Killer cytotoxic activity” which seems to be vague and difficult to understand.
The entire rationale should be part of the introduction and should not be stay distinct. It should be short, very technical, and should be an executive justification for why the study was conducted, and the problem hoped to be solved with some novel approach.

Experimental design

Materials and methods
No explanation was given for why the criteria were selected for eligibility, studies, data extraction, and quality assessment regions. For example, why was the Joanna Briggs Institute (JBI) critical assessment checklist used, of all checklists? Is it the only checklist that can be used? What is its advantage? Why were only 2 authors and 2 reviewers used? Why not more? Why not less? The questions should be answered in concise but technical terms in each section
Line 162 states “Two of the reviewers (S.B and H.A) WERE PERFORMED THE INCLUSION…” This grammar should be checked for soundness.

Validity of the findings

Discussion
Lines 225-250 are recommendable only for the justification aspect of the introduction. They do not discuss results obtained.
Excessive use of the expression “the current systematic review” could make the discussion monotonous.
The question of HOW? was sparingly answered all through the discussion. For example, the Tumour Necrosis Factor alpha mentioned in lines 236-237 was not explained in full detail regarding how it, and IL-6 are reduced by exercise. A brief but comprehensive biochemical mechanism of HOW IT HAPPENS between the input and output variables should be offered. This would make the discussion full of sound information, and not just a comparative declaration with past studies.

Limitation and Conclusion.
The use of WE should be completely removed and revised. For example, “We included published original articles…” should be revised to “Published original articles were included…”
Authors mentioned in lines 311-312 that “Understanding the mechanism between exercise and immune function in various cancer patients, as well as the relationship between these immunological markers and clinical outcomes, ARE ESSENTIAL FOR THE FUTURE STUDIES.” A basic search from google scholar and other scientific literature sources would show that several PAST studies have defined tis relationship to some extent. It is suggested that the relationships already published be mentioned in the discussion section.

Additional comments

The grammatical construction of the entire manuscript should be revised. For example, the excessive and unwarranted use of semi-colon ";".

---

## Round 0.2 · Minor Revisions

· Academic Editor

Minor Revisions

Dear Authors,

Two experts in the field reviewed your R1 version of the manuscript, indicating some minor points you should address.

Reviewer 2 ·

Basic reporting

I see a lot of improvement in this area.
Some suggestions:

Line 26: cancer does not only affect the body, but also has a psychological, societal and economic impact.
Line 27: The physical activity --> Physical activity (no need for The)

Line 129:
Inclusion criteria
- You should add that you only include studies that have a physical activity/exercise intervention and those who look at your outcome criteria. The inclusion criteria you actually mentioned in line 123-124. As of now, your list is not complete.

Line 201-208: For me, it would make more sense in you report on this before the actual findings from the studies (line 181-200).

Line 256: exercise-programs and not Programs

Table 2:
Your results and conclusion column is very similar. Maybe in the results column, you can work schematically and less with text and sentences. So it is very clear what your findings were and what findings were significant

Supplementary file 1 table 2:
Could you add the questions of the JBI in the legend of this table. Then the table can be read separately from the manuscript.

Experimental design

The research aim is now clear.

Methods:
Line 114:
You incorporated another database Sciencedirect on 15-02-2022. However, did you repeat your search strategy again in PubMed, Scopus and Cochrane? A check of these databases might show more relevant literature.

Results:
Overall, you could make a synthesis based on your three outcome categories: QoL, fatigue and immunity markers. And describe your results per study in those categories. Your table 2 gives an overview of each study, so within your result section, I want to find a synthesis and not just the same overview in text.
Also, were the outcome measures only measured once? What about follow-up? Was this reported in the studies, it might be relevant to know how these outcome measures change with time.

Validity of the findings

Discussion:
Line 224-226: You report on the findings on an increase in NK-cell cytotoxicity, could you elaborate on that. What does this mean for a cancer patient in treatment?
Line 224-227: You report on the findings on NK-cell cytotoxicity, but do not mention anything of your other outcome measures. Could you add very briefly what the most important findings were on all predefined outcome measures (Qol, fatigue, immunity markers) and then proceed with the rest of the discussion, where you go in depth on each outcome. Non-significant findings are also important findings.

Overall, I do miss a mention of limitations with your own review. The fact that you have such heterogenous studies (all types of cancer patients, all stages, all types of treatment, all ages, all types of physical activity …). Do you see any explanation why some studies did not find significant results? Maybe you can see that some types of physical activity seem to work better then others. This all stays quite general. You do mention that in future research, studies are needed that investigate the different forms of exercise. But perhaps, you already see some aspects emerge from your data.

Conclusion
This is very general. What does this systematic review specifically contribute to this topic?

Additional comments

I see many improvements in the basic reporting and methods area. However, a more in-depth discussion section will add a lot of value to this systematic review

·

Basic reporting

The revised manuscript is a greatly improved version of the previous one. The authors have shown mastery of the research procedure involved in the study and have made important rectifications. However, there are some minor suggestions that might further improve the scholarly appeal of the manuscript.

Experimental design

This section is satisfactory.

Validity of the findings

The conclusion for such an important study is quite short and nominal. The declaration that "Regular exercise has been shown to improve QOL, reduce depression and improve immunity
markers in cancer patients" is a quite generall known fact. The same principle is typically advised for a normal healthy lifestyle.

The question is, what is the unique finding and coclusion of the study? How are those unique findings going to affect the future? What steps should stakeholders (patients and health personnels) take to achieve better outcomes for cancer management?

These thoughts can be addressed in the conclusion section.

---

## Round 0.3 · Minor Revisions

· Academic Editor

Minor Revisions

Dear Authors,

Two experts in the field reviewed your manuscript. Some minor issues still persist that you should consider while revising the manuscript.

Reviewer 2 ·

Basic reporting

Still some issues with spelling or grammatical errors. So I would suggest to let a fluent English speaker read your work as a whole again.
Some examples:
Line 53: prescribe? You mean describe?
Line 226 these findings suggested --> suggest that: present tense is more suited
Line 295: a rehabilitation programs --> program : singular in stead of plural

Experimental design

No comment

Validity of the findings

Line 46-48 : Living an active lifestyle before diagnosis. Before diagnosis, cancer patients are healthy people. Can you make your recommendation on cancer patients while undergoing treatment or after treatment?

Line 332: Why don't you mention the heterogeneity of the included studies as a limitation and strength? Limitation as it is difficult to say anything about one specific type of cancer, but you give a broad overview of the cancer population as a whole.

·

Basic reporting

The revised manuscript has been greatly improved

Experimental design

This section is satisfactory.

Validity of the findings

This section is satisfactory.

---

## Round 0.4 · accepted · Accept

· Academic Editor

Accept

Dear Authors,
the changes you made meet the suggestions of all the reviewers.